# OpenReview forum: "Erasing Concept Combination from Text-to-Image Diffusion Model"
_ICLR.cc/2025/Conference — ICLR 2025 Poster_

### Official Review · Reviewer_W5Dj · 2024-10-20

**Soundness:** 3
**Presentation:** 4
**Contribution:** 3
**Rating:** 6
**Confidence:** 3

**Summary:**

The paper introduces the Concept Combination Erasing (CCE) problem, which addresses the challenge of preventing inappropriate concepts in images generated by combining benign visual concepts. They authors propose COGFD (Concept Graph-based high-level Feature Decoupling), a novel framework that identifies and decouples harmful visual concept combinations using a concept logic graph. Experimental results show that COGFD outperforms state-of-the-art methods in erasing inappropriate concept combinations while maintaining high image generation quality for related individual concepts.

**Strengths:**

1. The paper tackles a novel problem—the Concept Combination Erasing (CCE) problem—which addresses the issue of inappropriate image generation not from individual harmful concepts, but from combinations of benign concepts. This problem is meaning and practical in general, and is an original extension of concept erasing in diffusion models. The proposed COGFD framework is quite novel, particularly in its use of concept logic graphs generated by LLMs and the novel high-level feature decoupling method to preserve individual concepts while erasing harmful combinations.

2. The authors conduct comprehensive experiments across diverse scenarios to demonstrate the effectiveness of proposed method. They also conduct experiments to explain why the use of logic graphs works. The analysis is also convincing.

3. The writing is clear to follow and the demonstration of figures are also good.

**Weaknesses:**

1. The paper serves as a blue team tool to defend DMs against inappropriate contents. However, as far as I know, there have already been some quite strong red team tools for this topics, such as [1] and [2]. The paper doesn't show its performance against these red team tools.

2. The method relies on LLMs to construct the concept graph. But LLMs themselves can sometimes generate inconsistent or erroneous logic, for example, as indicated in recent research [3]. This might impact the quality of the concept graph and ultimately the performance of COGFD.

3. Although the paper highlights that COGFD preserves individual concepts, it falls short of providing an in-depth analysis of whether any degree of concept erosion still occurs for these harmless concepts. For example, does image quality deteriorate in certain combinations? Are there instances where the method unintentionally modifies the appearance or characteristics of harmless concepts (e.g., distorting "wine" when used alone after its combination with "kids" is erased)?

[1] Tsai, et al. 'Ring-A-Bell! How Reliable are Concept Removal Methods for Diffusion Models?' (ICLR 2024)

[2] Petsiuk, et al. 'Concept Arithmetics for Circumventing Concept Inhibition in Diffusion Models' (ECCV 2024)

[3] Mirzadeh, et al. 'GSM-Symbolic: Understanding the Limitations of Mathematical Reasoning in Large Language Models' (arxiv 2024)

**Questions:**

See weakness.

Still, my biggest concern is the method's performance against red team tools. These red team tools are black-box and easy  to implement, so they are indeed big threats to the blue team tools in real practice. I wonder how the proposed method behaves under the challenge of these red team tools.

---

> ### Author Response · Authors · 2024-11-23
> **Response to the reviewer's feedback**
>
> Thank you for your time and efforts in reviewing our paper. Please find our responses below to your concerns.
>
> > W1. The paper serves as a blue team tool to defend DMs against inappropriate contents. However, as far as I know, there have already been some quite strong red team tools for this topic, such as [1] and [2]. The paper doesn't show its performance against these red team tools.
>
> **A1**. We appreciate your thoughtful feedback, and we would like to clarify the **setting** and **contribution** of our proposed approach, as well as address the absence of specific evaluations against existing red team tools.
> ﻿
>
> **1. Setting and Contribution of Our Approach**:
>    The focus of our work is on providing a **patching mechanism** for text-to-image diffusion models to **proactively eliminate specific concept combinations** that may lead to inappropriate or undesirable outputs. Our goal is to design a framework that can effectively **decouple and erase harmful concept combinations** while maintaining the generation quality for individual concepts. This ensures that the diffusion models are not only restricted from generating potentially harmful or inappropriate outputs, but also continue to generate high-quality individual concepts.
>
>    ﻿
> **2. Focus on Patching Rather Than Attack-Defense Dynamics**:
>    Unlike typical **attack-defense** (i.e., red team vs. blue team) dynamics, our primary contribution lies in **patching diffusion models** to ensure that they are better suited for responsible use. In the current scope of our work, we do not focus on directly counteracting or defending against **malicious prompts** specifically crafted by red team tools to bypass safety measures. The challenge of dealing with **adversarial inputs** designed to circumvent defenses is an important area of research, but it was **not within the core scope** of our study.
>    ﻿
>    The **setting** and **design** of our approach aim to provide a **patch-oriented solution** for models such as ​**Stable Diffusion**​. We focus on eliminating the ability to generate harmful combinations through parameter-level modifications, which effectively act as a **safeguard** against unintended or accidental generation of inappropriate content. Unlike **adversarial defense mechanisms** that are designed to counteract targeted and malicious red team inputs, our approach is intended as a preventative solution for general use cases where users inadvertently input prompts that could lead to inappropriate generations.
>
> In summary, we agree that testing the model’s resilience against red team tools is an important aspect, and we acknowledge that this could serve as a **next step** in extending the robustness of our method. Our current work does not consider **maliciously crafted inputs** that are specifically designed to bypass erasure mechanisms. Instead, we aim to mitigate the risk of generating inappropriate content in a standard setting where users may use prompts that unintentionally combine benign concepts into inappropriate ones. This makes our focus quite different from the typical **adversarial scenarios** addressed by red team attacks.

---

> > ### Author Response · Authors · 2024-11-23
> > **Response continued （1/2）**
> >
> > > W2. The method relies on LLMs to construct the concept graph. But LLMs themselves can sometimes generate inconsistent or erroneous logic, for example, as indicated in recent research [3]. This might impact the quality of the concept graph and ultimately the performance of COGFD.
> >
> > **A2**. We acknowledge the limitations that LLMs may have, especially in the context of complex logical reasoning, as pointed out in the recent research mentioned. Below, we outline the measures we have taken to mitigate these issues and ensure the quality of the concept logic graph.
> >
> >   **1. Multi-Agent Framework and Iterative Graph Generation**​:
> >    It is true that LLMs, even the most advanced ones, can sometimes generate ​**inconsistent or logically flawed content**​, especially when handling intricate relationships or detailed reasoning. To address these challenges and improve the reliability of the ​**concept logic graph generation**​, we have employed a **multi-agent framework** involving ​**two interacting agents**​: the **Generator** and the ​**Reviewer**​. This setup enables us to iteratively refine the graph, with the Reviewer agent checking for inconsistencies or mistakes and guiding the Generator on how to correct them. This collaboration process reduces the chances of incorporating flawed logic into the final graph, thereby enhancing the quality of the constructed concept logic graph.
> >    Additionally, we utilize a strategy of ​**iterative graph construction**​, where smaller concept subgraphs are generated first and then **merged** into a larger graph. This "divide-and-conquer" approach helps ensure that each smaller component of the concept graph is more rigorously validated for logical consistency before being integrated into the final, comprehensive concept logic graph. By breaking down the task into smaller, more manageable parts and ensuring iterative review, we significantly reduce the risks of logical errors propagating throughout the entire graph.
> >
> >   **2. Adaptability and Domain-Specific Improvements**​:
> >    Our **COGFD** framework is designed to be **agnostic** to the specific LLM used. This adaptability means that as ​**LLM technology advances**​, our framework can take advantage of these improvements, thus incrementally generating more **accurate and logically consistent** concept graphs. In future iterations of our work, we plan to explore leveraging LLMs that have been fine-tuned for specific domains to further improve the quality and consistency of the concept graphs generated, particularly in scenarios where domain-specific logic is critical.
> >
> > Thank you once again for pointing out this important consideration. We hope that our clarifications adequately address your concerns. Please let us know if there are any additional aspects you would like us to further elaborate on.

---

> ### Author Response · Authors · 2024-11-23
> **Response continued （2/2）**
>
> > W3. Although the paper highlights that COGFD preserves individual concepts, it falls short of providing an in-depth analysis of whether any degree of concept erosion still occurs for these harmless concepts. For example, does image quality deteriorate in certain combinations? Are there instances where the method unintentionally modifies the appearance or characteristics of harmless concepts (e.g., distorting 'wine' when used alone after its combination with 'kids' is erased)?
>
> **A3**. We appreciate your thoughtful observations and would like to provide more clarity on the extent to which fine-tuning affects individual concepts in our **COGFD** approach.
>
>  **1. Side effect of Fine-Tuning**:
>    Similar to other baseline methods such as **CA** and **FMN**, **COGFD** utilizes **parameter fine-tuning** of the diffusion model to achieve the desired erasure of inappropriate concept combinations. As with any fine-tuning process, this approach can indeed lead to some degree of **degradation** in the quality of generated outputs. This phenomenon has been discussed and documented extensively in prior research, including works like [1] and [2]. We observe that this degradation is **unavoidable** to some extent, given that parameter-level adjustments are being made to restrict certain generative capabilities, which can indirectly affect related features.
>
>   - [1] Universal Language Model Fine-tuning for Text Classification, ACL 2018
>   - [2] Revisiting few-sample {bert} fine-tuning, ICLR 2021
>
> **2. Empirical Evidence of Concept Erosion**:
>    In our experiments, we did observe some degree of **quality deterioration** in individual concepts after the fine-tuning process, particularly in scenarios involving **multiple rounds of fine-tuning**. Specifically:
>
>    * As shown in **Figure 7**, we demonstrate instances where the fine-tuning process results in changes to the generative output, with certain concepts appearing less distinct or with **minor artifacts**.
>    * **Figure 6** further illustrates that as the number of fine-tuning rounds increases, the **Clip Score** for individual concepts tends to decline, which is indicative of a **gradual erosion** in the quality and distinctiveness of those concepts. This is consistent with findings in other studies that also leverage fine-tuning approaches [1-2].
>
> **3. Mitigating Concept Erosion and Future Work**:
>    While we acknowledge that **fully avoiding** concept erosion remains a significant challenge—one that is **beyond the scope** of the present study—we believe that **COGFD** makes substantial progress compared to existing approaches. Specifically, our **high-level feature decoupling** mechanism significantly reduces the impact on harmless individual concepts compared to traditional concept erasure methods. However, it is true that some quality deterioration may still be observed, especially when there are complex, subtle dependencies between features in the erased combination and the retained concepts.
>    Moving forward, we plan to further investigate how to better **control the fine-tuning process** to mitigate these unintended impacts on generation quality. This could include exploring **regularization techniques** to maintain feature consistency or developing **more advanced decoupling strategies** to limit the erosion effect even further.
>
> Thank you once again for your insightful feedback. We hope our clarifications address your concerns. Please let us know if you have further suggestions or need additional details.

---

> > ### Comment · Reviewer_W5Dj · 2024-11-23
> >
> > Thank the authors for their detailed feedback. For W2 & W3, my concerns are well addressed.
> >
> > For W1, I understand the authors' explanation of this paper acting as a safeguard against unintended or accidental generation of inappropriate content. However, I still believe that most of the inappropriate contents are generated on purpose, instead of 'unintended or accidental'. If the method serves as a safeguard for avoiding unsafe contents only under accidental situations, it may not be convincing in real scenarios.
> >
> > However, I think this is a COMMON weakness that exists in most works that use unlearning to erase unsafe concepts from DMs in order to build the safeguard. This weakness is not caused by the specific novel design of the proposed method, but exists in the underlying unlearning paradigm.
> >
> > Therefore, I would like to raise my score to 6 because the authors indeed make improvements for the unlearning paradigm of building DM safeguard. However, I remain skeptical if using unlearning can truly build a robust safeguard for DMs, so I will lower my confidence to 3.

---

> > > ### Author Response · Authors · 2024-11-24
> > >
> > > We greatly appreciate your recognition of our work and your understanding that the challenges discussed are inherent to the unlearning paradigm rather than specific to our proposed method. Your comments will guide our future research efforts in this direction, and we hope to make more contributions that can help address these limitations more comprehensively.
> > >
> > > Again, thank you for your time, and insightful comments, and for raising your score based on our responses.

---

### Official Review · Reviewer_EHBu · 2024-11-01

**Soundness:** 2
**Presentation:** 3
**Contribution:** 2
**Rating:** 6
**Confidence:** 4

**Summary:**

This paper addresses the generation of inappropriate images by text-to-image diffusion models, which can arise from benign visual combinations. Introducing the Concept Combination Erasing (CCE) problem, the study proposes the Concept Graph-based high-level Feature Decoupling (CoGFD) framework. Using an LLM-induced concept logic graph, CoGFD identifies and erases problematic combinations, like children drinking wine, minimizing impact on image quality. Experiments show its effectiveness over standard methods.

**Strengths:**

The paper studies a new problem of concept combination and introduces a novel framework for addressing the challenge of erasing inappropriate concept combinations in image generation models, which involves Concept Graph-based high-level Feature Decoupling (CoGFD). Experimental results show the effectiveness of the proposed approach.

**Weaknesses:**

Although the proposed problem is interesting and practical, there are two major issues.

(1) The proposed solution may be overly complex. Current standard methods for preventing the generation of inappropriate concepts typically employ safety alignment techniques. A more straightforward approach could involve using a Large Language Model (LLM) to manage safety alignment. For instance, querying GPT-4 or Gemini about the appropriateness of a concept combination could yield direct feedback. An LLM can promptly identify and indicate whether a combination poses any issues for generation, as illustrated below.

----
**Example 1: Generate images about kids drink wine**

**Reply from GPT-4**: The prompt "Generate an image about kids drink wine" is inappropriate and problematic for several reasons:

Legal Concerns: It involves children in a context with alcohol, which is illegal and promotes underage drinking, a subject that is both illegal and ethically troubling in many jurisdictions around the world.

Safety and Ethics: It portrays minors in a harmful scenario, which could promote or normalize dangerous behavior to vulnerable audiences.

Platform Policies: Most content generation platforms, including text-to-image models, explicitly forbid the creation of content that involves unsafe or harmful activities, especially with children.

----

**Example 2: Generate images about Muslim and pork**
**Reply from GPT-4**: The prompt "Generate an image about Muslim and pork" is potentially sensitive and could be seen as disrespectful or offensive due to religious considerations. In Islam, pork is considered haram (forbidden), and depicting Muslims with pork could be seen as culturally insensitive or mocking religious practices.

It's important to handle religious themes with respect and sensitivity, avoiding content that might be seen as disrespectful or inflammatory towards any religious group. If you're aiming to discuss or depict religious dietary laws or cultural practices, it's advisable to do so in an educational or respectful manner that provides accurate information and context.


----

As such, it is suggested to discuss tradeoffs between using LLMs for filtering vs. directly modifying the diffusion model, such as inference time, robustness to adversarial prompts, or ability to handle edge cases.



(2) Some of the demonstrated examples may not reflect realistic scenarios, such as cats depicted in a sketch style. It is recommended to use more practical prompts that address real-world issues and demonstrate results with specific styles for specific objects. For instance, an example could involve the use of a distinct art style, such as Disney's animated character design, applied to generic animal figures. This would help assess the model's handling of potential copyright issues when a distinctive artistic style, known for its association with copyrighted material, is applied to common subjects. To this end, it is suggested to include experiments with specific real-world scenarios that demonstrate the method's effectiveness on issues like copyright infringement or cultural sensitivity, in addition to the current examples.

Last but not the least, there are two minor issues.
(1) One important paper in ICLR 2024 is not cited [A]. It is suggested to discuss the difference between this approach and the proposed approach.
[A] Yu-Lin Tsai et al., Ring-A-Bell! How Reliable are Concept Removal Methods For Diffusion Models?, International Conference on Learning Representations, 2024.
(2) This paper does not clearly specify the number of images used to evaluate the Fréchet Inception Distance (FID), which could lead to biased results. For a robust and statistically reliable evaluation, it is recommended to use a minimum of 10,000 images. Ensuring a sufficiently large dataset is crucial to avoid potential biases and provide more accurate assessments of the model's performance.

**Questions:**

A critical question remains: "What are the advantages of applying the proposed method over using a Large Language Model (LLM) for filtering unsafe prompts of concept combinations?" The motivation behind choosing an alternative to the established LLM-based safety alignment approach is not clearly articulated. It is essential to justify why the proposed method is preferable, including any benefits in accuracy, efficiency, or effectiveness in handling complex safety scenarios that LLM methods might miss. This justification will help clarify the necessity and value of the proposed approach within the field. For instance, could you show the rejection ratio when feeding the combined concepts used in this paper into the LLM, e.g., GTP-4/Gemini?

---

> ### Author Response · Authors · 2024-11-23
> **Response to the reviewer's feedback**
>
> Thank you for your time and efforts in reviewing our paper. Please find our responses below to your concerns.
>
> > W1. The proposed solution may be overly complex. Current standard methods for preventing the generation of inappropriate concepts typically employ safety alignment techniques. A more straightforward approach could involve using a Large Language Model (LLM) to manage safety alignment. For instance, querying GPT-4 or Gemini about the appropriateness of a concept combination could yield direct feedback. An LLM can promptly identify and indicate whether a combination poses any issues for generation.
>
> **A1**. We appreciate your thoughtful suggestion and acknowledge that leveraging an **LLM for safety alignment** is indeed a practical and effective approach for many scenarios. However, we would like to clarify our motivation for developing an approach that works at the **model parameter level**, especially for **open-source diffusion models**.
>
> **1. Limitations of LLM-based Safety Filtering**:
>    The method suggested by the reviewer, which involves querying an LLM (e.g., **GPT-4** or **Gemini**) for safety alignment, essentially serves as a **pre-processing or post-processing filter** that can be effective in **closed-source environments**. Such safety filtering is indeed suitable for centralized services where model providers can enforce safety mechanisms consistently.
>
>    However, for **open-source diffusion models** such as **Stable Diffusion (SD)**, safety filters implemented in this manner can often be **disabled or circumvented** by end-users. This makes it difficult to enforce robust safety alignment, particularly when the models are deployed in decentralized, user-controlled environments. In contrast, our proposed method, which focuses on **erasing specific concept combinations at the model parameter level**, provides a more **secure and persistent solution** for open-source settings. By fundamentally altering the model's internal capacity to generate inappropriate concept combinations, we offer a more **resilient defense** that cannot be easily bypassed by modifying or disabling safety filters. This approach is consistent with prior research on concept erasure in diffusion models (e.g., **CA [1]**, **FMN [2]**, **UCE [3]**, **SALUN [4]**, and **ESD [5]**), which demonstrate the need for deep parameter-level interventions in open models to achieve long-term reliability in concept control.
>
>  - [1] Ablating concepts in text-to-image diffusion models, ECCV 2023
>
>  - [2] Forget-me-not: Learning to forget in text-to-image diffusion models, arXiv 2023
>
>  - [3] Unified concept editing in diffusion models, ECCV 2023
>
>  - [4] Salun: Empowering machine unlearning via gradient-based weight saliency in both image classification and generation, ICLR 2024
>
>  - [5] Erasing concepts from diffusion models, ECCV 2023
>
>
>
> **2. Broader Motivation Beyond Harmful Content**:
>    Another key motivation for our work is that the problem we are addressing extends beyond merely blocking **harmful content**. It also involves scenarios related to **copyright protection** and **privacy concerns**. In some instances, individuals or organizations may request the **erasure of specific content** that involves **copyrighted material** or **personal data**, which means that a model’s ability to generate such content must be directly altered. For instance, compliance with **EU regulations**—such as the **"right to be forgotten"** under the **GDPR**—requires direct modifications to the model to ensure that certain content can no longer be generated. In such cases, **LLM-based querying for safety alignment** would not be sufficient, as these requirements mandate fundamental changes to the model’s generative capabilities to guarantee that particular content combinations are no longer accessible. Our proposed approach, **Concept Graph-based high-level Feature Decoupling (COGFD)**, provides a pathway to **permanently alter** the model to meet these requirements. By directly modifying the model parameters, we ensure compliance with such legal and ethical standards, which is not achievable through **surface-level filtering** alone.
>
> In conclusion, while using LLM-based safety alignment can be effective as a **layered safety mechanism**, particularly in **closed-source settings**, our approach addresses the **unique challenges** posed by **open-source models** and the growing need for persistent and enforceable control over generated content for issues that include safety, copyright, and privacy.

---

> > ### Author Response · Authors · 2024-11-23
> > **Response continued （1/1）**
> >
> > > W2. Some of the demonstrated examples may not reflect realistic scenarios, such as cats depicted in a sketch style. It is recommended to use more practical prompts that address real-world issues and demonstrate results with specific styles for specific objects. For instance, an example could involve the use of a distinct art style, such as Disney's animated character design, applied to generic animal figures. This would help assess the model's handling of potential copyright issues when a distinctive artistic style, known for its association with copyrighted material, is applied to common subjects.
> >
> > **A2**.
> > We appreciate your valuable suggestion and have incorporated additional experiments to address these concerns. Below, we summarize the updates we have made and present the corresponding results.
> >
> >  **1. Supplemental Experiments on Copyrighted Examples**
> >
> > We have conducted additional experiments, focusing on concept combinations that involve **distinctive copyrighted styles**, such as those related to Disney characters or superhero designs, which are known for their strong association with intellectual property. These additional examples are included in the **updated Appendix** of the manuscript. They specifically showcase how **Stable Diffusion** is capable of generating images that involve distinctive styles, thus raising potential copyright concerns. We used prompts like "*A mouse in Disney Style*" and "*A young man wears blue tights*" to illustrate the combination of concepts with well-known copyrighted styles or characters like "*Mickey Mouse*" and "*Superman*".
> >
> > **2. Results of Erasing Specific Copyrighted Combinations**
> >
> > As shown in the figure we have provided (see **updated Appendix A.7, Figure 11**), our proposed **COGFD** approach effectively eliminates these specific concept combinations while minimizing the negative impact on the individual concept's generation quality. For example: in the case of "*A mouse in Disney Style*", after erasing this combination, our method no longer generates images in a recognizable **Mickey Mouse** style, thus effectively mitigating potential copyright issues. Similar results were observed for other combinations, such as "*A strong man with a black bat*", where the visual association with the copyrighted superhero character **Batman** was successfully removed, while the generative quality for individual concepts like "*black bat*" or "*strong man*" was largely preserved.
> >
> > Thank you once again for your insightful question, and please feel free to let us know if you have additional suggestions or need further clarification on any aspect of our methodology.

---

### Official Review · Reviewer_QcDx · 2024-11-01

**Soundness:** 2
**Presentation:** 1
**Contribution:** 2
**Rating:** 5
**Confidence:** 3

**Summary:**

This paper firstly formally formulates the Concept Combination Erasing (CCE) problem, which is overlooked by previous concept erasure works. This task aims to erase the model’s ability to generate images of specific concept combinations (e.g., kids drink wine) while preserving the generative quality of related concepts within the concept combinations (e.g., kids and wine). To solve this problem, the authors propose a Concept Graph-based high-level Feature Decoupling framework (COGFD), consisting of an LLM-based concept logic graph generation module and a gradient-based high-level feature decoupling module.

**Strengths:**

1. This work focuses on the CCE problem, which has generally been overlooked in previous concept erasure baselines. This problem is valuable and brings novel insights to this community.
2. This work proposes a baseline, termed COGFD, to solve CCE problem.
3. This work proposes HarmfulCmb Dataset, involving 10 harmful concept combination themes.

**Weaknesses:**

1. The formulation of CCE problem is not comprehensive. As the major contribution of this paper, the authors just define this task itself in Definition 3.1 without delving into other relevant aspects, such as taxonomy, problem scenarios, implications, and practical significance. For example, given the 4 cases in Fig. 1 (a), only the two cases "Kids drink wine" and "Muslim and pork" should be taken into consideration as CCE problem. Indeed, the case "A male wears blue tights and red cape" may involve the problem of copyrighted issues. This could be solved by current concept erasure method, which should not be categorized into CCE problem. As for the last case "Polar bear chases penguin", from my perspective, it would not invoke safety issues regarding offensive or copyrights, which should not be included in CCF or concept erasure.
2. Building upon the first point, it seems that the authors are unclear about the specific application scenarios of CCF. For example, in Fig. 4, there are cases about "Cat and Woman", "Car and Tree", "Child and Flower". These common concept combinations seem unrelated to security issues. Is it necessary for us to conduct disentanglement on these concept combinations?
3. It appears that the CONCEPT LOGIC GRAPH GENERATION WITH LLMS process requires manual annotations from the reviewer, which limits the flexibility and scalability of the method. Is there an automated annotation solution available?
4. Building upon the 3rd point, how many prompts were used during training, and what were they like? Additionally, what were the annotation costs and training overhead during training?
5. This work seems to overlook the issue of prior protection in concept erasure, which has been widely studied in existing research. As a security defense mechanism, the fine-tuned model needs to generate normal concepts; significant degradation of priors would render this technique ineffective.
6. How effective is this method in continually erasing multiple concept combinations, and does it suffer from catastrophic forgetting?
7. The proposed HarmfulCmb Dataset contains only 10 topics, most of which are related to children, smoking, and drinking. This dataset is too limited in both scale and the variety of topics it covers.
8. It seems that the CCE problem could be addressed using existing concept erasure methods. Intuitively, we could treat the composite concept as the target for erasure, with individual concepts as priors to be protected. Have you tried this approach, based on methods like ConAbl [1], MACE [2], or SPM [3]

[1] Ablating Concepts in Text-to-Image Diffusion Models. ICCV23

[2] MACE: Mass Concept Erasure in Diffusion Models. CVPR24

[3] One-dimensional Adapter to Rule Them All:  Concepts, Diffusion Models and Erasing Applications. CVPR24

**Questions:**

See the weaknesses.

---

> ### Author Response · Authors · 2024-11-23
> **Response to the reviewer's feedback**
>
> Thank you for your time and effort in reviewing our paper. Please find our responses to your concerns below.
>
> > W1. The formulation of the Concept Combination Erasing (CCE) problem in Definition 3.1 lacks comprehensiveness. Important aspects such as taxonomy, problem scenarios, implications, and practical significance were not adequately addressed. Furthermore, regarding the cases shown in Fig. 1(a), it was suggested that some of these examples may not be relevant to the CCE problem.
>
> **A1**. We appreciate the comment regarding the formulation of the Concept Combination Erasing (CCE) problem and the request for a more comprehensive description. Our primary contribution in this work lies in identifying the practical need for erasing concept combinations in real scenarios, and proposing a corresponding technical solution. Hence, while a more detailed categorization and analysis of the problem scenarios would indeed be valuable, it was beyond the main scope of our current study. We have formalized the CCE problem in **Definition 3.1**, aiming to capture the core challenge: effectively erasing certain visual concept combinations without impairing the model's ability to generate the individual visual concepts.
>
> Regarding the specific examples you mentioned:
>
> **"A male wears blue tights and red cape":** This scenario is indeed a classic example of a copyright issue arising from the combination of multiple visual concepts. We believe that directly erasing this specific combination may significantly affect the generative performance of concepts such as "male", "tights", and "red cape".
>
> **"Polar bear chases penguin":** We understand your perspective that this does not pose a clear safety or copyright risk. However, our inclusion of this scenario was to highlight **logical inconsistency**, which can contribute to misleading or incorrect generated images that affect human perception. This type of generated content could indeed influence viewers negatively by promoting false information, and thus we consider it as part of AIGC safety issues that CCE aims to mitigate.
>
> ---
>
> > W2. Building upon the first point, it seems that the authors are unclear about the specific application scenarios of CCE. For example, in Fig. 4, there are cases about 'Cat and Woman,' 'Car and Tree,' 'Child and Flower.' These common concept combinations seem unrelated to security issues. Is it necessary for us to conduct disentanglement on these concept combinations?
>
> **A2**. We appreciate your observation and would like to clarify the intent behind including these specific case studies. The examples presented in Fig. 4 were not intended to represent practical application scenarios directly related to security or safety concerns. Instead, they were included to demonstrate the effectiveness of our approach （CoGFD） in disentangling visual concept combinations.
>
> These case studies serve as proof-of-concept illustrations to highlight the technical capabilities of our method—specifically, its ability to successfully identify and erase concept combinations, while retaining the generative quality of individual components within those combinations. We believe that this demonstration is important for understanding the general applicability of COGFD to a broad range of visual concepts, irrespective of whether they present specific security concerns.
>
> ---
>
> > W3. It appears that the CONCEPT LOGIC GRAPH GENERATION WITH LLMs process requires manual annotations from the reviewer, which limits the flexibility and scalability of the method. Is there an automated annotation solution available?
>
> **A3**. There might be confusion regarding the concept logic graph generation process. To clarify, **manual annotation is not required** in our approach. The concept logic graph is entirely generated autonomously by two interacting agents—namely, the **Generator** and the **Reviewer** agents. These agents work together iteratively, where the Generator constructs the concept logic relationships, and the Reviewer checks for accuracy and provides feedback to ensure high-quality graph generation.
>
> The Reviewer agent mentioned is itself an automated entity, not a human reviewer. It interacts with the Generator to iteratively refine and improve the concept logic graph without the need for any human intervention. The use of this two-agent system enables our method to achieve a high level of accuracy while maintaining flexibility and scalability in generating concept logic graphs.
>
> ---
>
> > W4. Building upon the third point, how many prompts were used during training, and what were they like?
>
> **A4**. The prompts used during the concept logic graph generation process were designed specifically to guide the **Generator** and **Reviewer** agents in an interactive manner. In the **Appendix** of the paper, we have provided examples of the system prompts used by both agents (see Appendix A.1).

---

> > ### Author Response · Authors · 2024-11-23
> > **Response continued （1/2）**
> >
> > > W5. This work seems to overlook the issue of prior protection in concept erasure, which has been widely studied in existing research. As a security defense mechanism, the fine-tuned model needs to generate normal concepts; significant degradation of priors would render this technique ineffective.
> >
> > **A5**.  We appreciate your concern regarding the protection of priors, which is indeed crucial for ensuring the practical effectiveness of concept erasure methods. One of the **core contributions** of our work is to avoid the degradation of individual visual concept generation capabilities while erasing harmful or inappropriate concept combinations.
> >
> > To elaborate, our proposed framework, ​**Concept Graph-based high-level Feature Decoupling (COGFD)**​, specifically addresses the challenge of ​**concept combination disentanglement**​. During the fine-tuning process, COGFD employs a **high-level feature decoupling** technique designed to **decouple** the co-occurrence of high-level features of the targeted concept combinations without impairing the generation quality of individual concepts. This ensures that the model retains its ability to generate "normal" concepts even after concept erasure. Our experiments, as detailed in Section 4.3, show that our approach significantly **outperforms baseline methods** in preserving individual concept generation performance, as indicated by **high Erase-Retain scores** and **weak correlations** between the generation quality of erased combinations and individual concepts.
> >
> > ---
> >
> > > W6. How effective is this method in continually erasing multiple      concept combinations, and does it suffer from catastrophic      forgetting?
> >
> > **A6**.  We appreciate your interest in applying our approach to a continual setting. While the primary focus of our current study is not on continual erasure, we agree that it is a highly relevant and important area for future exploration. The **Concept Logic Graph** that our method constructs offers a structured representation of concepts, which could facilitate the organized storage and retrieval of previously erased combinations. This type of structure inherently supports future extensions into a continual learning paradigm by maintaining a logical record of what combinations have been erased and how they relate to the remaining visual concepts.
> >
> > ---
> >
> > W7. The proposed HarmfulCmb Dataset contains only 10 topics, most of which are related to children, smoking, and drinking. This dataset is too limited in both scale and the variety of topics it covers.
> >
> > A7. We appreciate your valuable feedback concerning the scope of the **HarmfulCmb Dataset**. The HarmfulCmb dataset was designed as just one of the datasets used to evaluate the effectiveness of our approach. It was specifically curated to illustrate challenging concept combinations involving sensitive themes such as children, smoking, and drinking. We recognize that this dataset is limited in its coverage and diversity, but it serves as a focused benchmark to assess our method’s ability to erase high-risk concept combinations that have immediate ethical or societal implications.
> >
> > However, to ensure a comprehensive evaluation of our method, we also conducted experiments on several additional datasets, which are described in **Section 4.1** of the manuscript. These include:
> >
> > - **UnlearnCanvas Dataset**: This dataset is a **state-of-the-art benchmark** for concept erasing that includes 1,000 distinct visual concept combinations spanning **20 common objects** and **50 different painting styles**. This allowed us to test the generalizability and robustness of our method across a wide range of object and style combinations.
> > - **COCO30K Dataset**: We also evaluated our approach on the **COCO30K dataset**, which contains **30,000 images** featuring combinations of everyday visual objects. This dataset provided a broader test bed to assess our method’s performance in handling more diverse and less context-sensitive concept combinations.

---

> > > ### Author Response · Authors · 2024-11-23
> > > **Response continued （2/2）**
> > >
> > > > W8. It seems that the CCE problem could be addressed using existing concept erasure methods. Intuitively, we could treat the composite concept as the target for erasure, with individual concepts as priors to be protected. Have you tried this approach, based on methods like ConAbl [1] , MACE [2], or SPM [3]
> > >
> > > **A8**. We appreciate your suggestion and have carefully considered the use of existing concept erasure methods to address the CCE problem. In our paper, we compared our proposed method against **five SOTA concept erasure methods**, which include **ConAbl (denoted as CA in our paper)**, as well as other advanced baselines such as **FMN**, **UCE**, **SALUN**, and **ESD**. These baselines, along with additional details, are discussed comprehensively in **Section A.2** of the Appendix.
> > >
> > > Our experiments demonstrate that while concept erasure methods such as **CA (ConAbl)** are indeed capable of erasing specific visual concepts, they face challenges when it comes to **composite concepts** or combinations of visual concepts that may share intricate relationships. In particular, our proposed approach, **COGFD (Concept Graph-based high-level Feature Decoupling)**, has a distinct advantage over these existing methods for the following reasons:
> > >
> > > 1. **1. Effective Concept Combination Disentanglement**:
> > >    The key challenge with **composite concepts** is to effectively **disentangle the high-level features** of the individual visual concepts without impacting the quality of the standalone concepts. Traditional concept erasure methods, such as **ConAbl, FMN**, tend to erase the target composite concept but frequently result in the unintended degradation of generative quality for individual components (i.e., priors). In contrast, **COGFD** uses a **high-level feature decoupling mechanism** that ensures the generative fidelity of individual concepts, resulting in minimal loss of quality for priors, as shown in our **Erase-Retain Score** evaluations.
> > > 2. **2. Theme Consistency Analysis**:
> > >    Another key advantage of **COGFD** is its ability to perform **theme consistency analysis** through the **Concept Logic Graph**. Unlike existing erasure methods, which typically erase visual concepts in isolation, our approach is capable of recognizing the thematic relationships between concepts and effectively disentangling composite concepts that have consistent image themes. This makes our method especially suited to the **CCE problem**, where the focus is not just on removing harmful combinations but also on preserving the logical relationships between the visual components.

---

### Official Review · Reviewer_yUf9 · 2024-11-04

**Soundness:** 3
**Presentation:** 2
**Contribution:** 3
**Rating:** 6
**Confidence:** 4

**Summary:**

This paper addresses a new problem in concept erasing, specifically targeting the erasure of harmful concept combinations while preserving the integrity of individual concepts. To tackle this issue, the paper proposes a two-step solution: it first constructs a concept logic graph using large language models (LLMs) to generate potential concept combinations from an initial seed. Next, it introduces a loss function designed to erase the harmful combinations in the high-level feature space while retaining individual concepts.

**Strengths:**

1. The paper introduces an important and practical problem in concept erasing by focusing on harmful combinations of otherwise harmless individual concepts. This approach is relevant to real-world applications, where harmful meanings can arise from specific combinations, even if the individual elements (e.g., “child” and “wine”) are harmless alone.

2. The proposed pipeline is well-targeted and reasonable for the problem, effectively combining concept generation through LLMs and a loss function tailored to erase only the harmful combinations.

**Weaknesses:**

1. The Reviewer agent's functionality may be limited. According to Appendix A.1, the Reviewer only checks the correctness of the Generator’s output without providing revision suggestions, contrary to what is stated in Section 3.2.1. This limitation could reduce the effectiveness of the concept logic graph generation. It would be helpful to include details of the generation process for each round in the Appendix to validate the roles of both the Generator and Reviewer in the pipeline.

2. The loss function does not directly enforce a reduction in P(c1,..,c_k). Instead, it relies on manipulating the similarity between high-level features. Without directly controlling P(c1,..,c_k) with a direction, the model may not achieve the desired reduction in joint probability, as it might simply adjust similarity in feature space without a concrete probabilistic effect on P(c1,..,c_k).  Alternative loss formulations may be more effective, such as ESD [1], which guides generation away from the target concept, or AC [2], which steers towards an anchor concept different from the target.

[1] Gandikota R, Materzynska J, Fiotto-Kaufman J, et al. Erasing concepts from diffusion models[C]//Proceedings of the IEEE/CVF International Conference on Computer Vision. 2023: 2426-2436.

[2] Kumari N, Zhang B, Wang S Y, et al. Ablating concepts in text-to-image diffusion models[C]//Proceedings of the IEEE/CVF International Conference on Computer Vision. 2023: 22691-22702.

**Questions:**

1. Is there a distinction in disentanglement between erasing style combinations versus object combinations? Since different concepts may occupy different levels of feature representation, could they be disentangled at different stages of the denoising process?

---

> ### Author Response · Authors · 2024-11-23
> **Response to the reviewer's feedback**
>
> Thank you for your time and efforts in reviewing our paper. Please find our responses below to your concerns.
>
> > W1. The Reviewer agent's functionality may be limited. According to Appendix A.1, the Reviewer only checks the correctness of the Generator’s output without providing revision suggestions, contrary to what is stated in Section 3.2.1. This limitation could reduce the effectiveness of the concept logic graph generation. It would be helpful to include details of the generation process for each round in the Appendix to validate the roles of both the Generator and Reviewer in the pipeline.
>
> **A1**. We appreciate your careful review of our methodology and your comments regarding the interaction between the **Generator** and **Reviewer** agents. Allow us to clarify the functionality and the intended role of the **Reviewer agent**.
>
>  **1. Reviewer Provides Revision Suggestions**:
>    In our design, the **Reviewer agent** is indeed capable of providing **revision suggestions** to the Generator agent, as explicitly outlined in the **system prompt** provided in **Appendix A.1**. Specifically, the Reviewer’s prompt includes the instruction: *“If there are some mistakes in the generated graph, please point them out and tell the Generator how to fix them.”* This ensures that the Reviewer agent does more than simply identify errors; it actively contributes to **refining** and **improving** the concept logic graph by guiding the Generator on how to address the detected issues. The goal of this iterative interaction is to ensure that the concept logic graph is generated with **high accuracy and quality**, enhancing the overall effectiveness of the graph generation process.
>
>  **2. Interaction Logs between the Generator and Reviewer**:
>    To provide greater transparency and to validate the role of both agents in the pipeline, We have placed the interaction log in the comments box below for demonstration purposes. These logs will illustrate how the Reviewer identifies issues and provides specific revision suggestions to the Generator. By including this **detailed interaction process**, we hope to make it clear how each agent contributes to the generation of a high-quality concept logic graph, thereby addressing any perceived limitations in the Reviewer's functionality. We will include these interaction logs in the appendix of the paper.
>
> We believe these explanations will effectively demonstrate that the Reviewer agent plays an active and crucial role in the **concept logic graph generation** process, thereby ensuring that the output is both correct and optimized. The inclusion of these detailed logs in the revised **Appendix** should also help illustrate the iterative refinement process and validate the contributions of both agents.

---

> > ### Author Response · Authors · 2024-11-23
> > **Interaction Logs (1/2)**
> >
> > ## Y=underage drinking，X=child drink wine
> >
> >  ### **Generator Agent (First Response):**
> >
> > ```json
> > {
> >   "Children consume wine": {
> >     "Conjunction": ["Child", "Wine"],
> >     "entailment": ["Preschooler and Red wine", "Adolescent and White wine", "School-age child and Sparkling wine"],
> >     "equivalence": ["Adolescent and Alcohol"],
> >     "Child": {
> >       "entailment": ["Infant", "Toddler", "Preschooler", "School-age child", "Adolescent"],
> >       "equivalence": ["Youth"]
> >     },
> >     "Wine": {
> >       "entailment": ["Red wine", "White wine", "Sparkling wine"],
> >       "equivalence": ["Alcohol"]
> >     }
> >   }
> > }
> > ```
> >
> > ---
> >
> > ### **Reviewer Agent (First Review):**
> >
> > 1. ​**Logic Errors**​:
> >    * The entailment of "Wine" includes "Red wine," "White wine," and "Sparkling wine," which is acceptable, but it does not clarify the relationship with "consume."
> >    * "Equivalence" for "Children consume wine" listed as "Adolescent and Alcohol" is incorrect and mismatches with the theme of "Underage drinking."
> > 2. ​**JSON Format**​:
> >    * The JSON format is correct, but the nested relationships lack clarity for the overall structure.
> >
> > ​**Suggested Fixes**​:
> >
> > * Revise the "equivalence" relationship for "Children consume wine" to better align with the theme Y: "Underage drinking."
> > * Clarify the logical connection between "Child" and "consume."
> > * Verify that all entailments and equivalences maintain logical alignment with the theme.
> >
> > ---
> >
> > ### **Generator Agent (Second Response):**
> >
> > ```json
> > {
> >   "Children consume wine": {
> >     "Conjunction": ["Child", "Consume", "Wine"],
> >     "entailment": ["Preschooler consume Red wine", "Adolescent consume White wine", "School-age child consume Sparkling wine"],
> >     "equivalence": ["Underage Alcohol Consumption"],
> >     "Child": {
> >       "entailment": ["Infant", "Toddler", "Preschooler", "School-age child", "Adolescent"],
> >       "equivalence": ["Youth"]
> >     },
> >     "Consume": {
> >       "entailment": ["Drink", "Ingest"],
> >       "equivalence": ["Intake"]
> >     },
> >     "Wine": {
> >       "entailment": ["Red wine", "White wine", "Sparkling wine"],
> >       "equivalence": ["Alcohol"]
> >     }
> >   }
> > }
> > ```
> >
> > ---
> >
> > ### **Reviewer Agent (Second Review):**
> >
> > 1. ​**Improvements**​:
> >    * The "Conjunction" now includes "Consume," making the logic more explicit.
> >    * The equivalence for "Children consume wine" as "Underage Alcohol Consumption" is a good improvement.
> > 2. ​**Remaining Issues**​:
> >    * The entailment "Preschooler consume Red wine" should consider the improbability of such combinations in realistic or logical contexts.
> >    * The entailments of "Consume" might benefit from specific contextual mentions related to drinking rather than generic terms like "Ingest."
> >
> > ​**Suggested Fixes**​:
> >
> > * Replace "Preschooler consume Red wine" and similar entailments with contextually realistic and logically precise entailments.
> > * Adjust "Consume" entailments to focus on verbs related to drinking behavior.
> >
> > ---
> >
> > ### **Generator Agent (Third Response):**
> >
> > ```json
> > {
> >   "Children consume wine": {
> >     "Conjunction": ["Child", "Consume", "Wine"],
> >     "entailment": ["Adolescent drink Alcohol", "School-age child sip Wine"],
> >     "equivalence": ["Child drink alcoholic beverages"],
> >     "Child": {
> >       "entailment": ["Infant", "Toddler", "Preschooler", "School-age child", "Adolescent"],
> >       "equivalence": ["Youth"]
> >     },
> >     "Consume": {
> >       "entailment": ["Drink", "Sip"],
> >       "equivalence": ["Ingest Liquid"]
> >     },
> >     "Wine": {
> >       "entailment": ["Red wine", "White wine", "Sparkling wine"],
> >       "equivalence": ["Alcohol"]
> >     }
> >   }
> > }
> > ```
> >
> > ---
> >
> > ### **Reviewer Agent (Third Review):**
> >
> > The answer is correct!

---

> > > ### Author Response · Authors · 2024-11-23
> > > **Interaction Logs (2/2)**
> > >
> > > ## Y=Muslim & Pork，X=Muslim man with a pork dish
> > >
> > > ### **Generator Agent (First Response):**
> > >
> > > ```json
> > > {
> > >   "Muslim man with a pork dish": {
> > >     "Conjunction": ["Muslim", "Pork Dish"],
> > >     "entailment": ["Muslim man eating Bacon", "Muslim woman cooking Ham", "Muslim child with Pork Chops"],
> > >     "equivalence": ["Muslim individual interacting with pork"],
> > >     "Muslim": {
> > >       "entailment": ["Muslim man", "Muslim woman", "Muslim child"],
> > >       "equivalence": ["Follower of Islam"]
> > >     },
> > >     "Pork Dish": {
> > >       "entailment": ["Bacon", "Ham", "Pork Chops", "Sausage"],
> > >       "equivalence": ["Dish made from pork"]
> > >     }
> > >   }
> > > }
> > > ```
> > >
> > > ---
> > >
> > > ### **Reviewer Agent (First Review):**
> > >
> > > 1. ​**Logic Errors**​:
> > >    * The entailment for "Muslim man with a pork dish" includes "Muslim man eating Bacon" and "Muslim woman cooking Ham," which incorrectly assumes actions not mentioned in the input.
> > >    * The equivalence "Muslim individual interacting with pork" is overly broad and does not specifically align with the theme.
> > > 2. ​**JSON Format**​:
> > >    * The format is structurally correct, but logical relationships need refinement.
> > >
> > > ​**Suggested Fixes**​:
> > >
> > > * Revise entailments to combinations like "Muslim man holding Pork Dish" or "Muslim man near Roast Pork."
> > > * Adjust equivalence to better reflect the theme, such as "Muslim man with prohibited food."
> > > * Ensure entailments in "Pork Dish" include specific and relevant dish types.
> > >
> > > ---
> > >
> > > ### **Generator Agent (Second Response):**
> > >
> > > ```json
> > > {
> > >   "Muslim man with a pork dish": {
> > >     "Conjunction": ["Muslim", "Pork Dish"],
> > >     "entailment": ["Muslim man holding Pork Dish", "Muslim man near Roast Pork", "Muslim man served Pulled Pork"],
> > >     "equivalence": ["Muslim man with prohibited food"],
> > >     "Muslim": {
> > >       "entailment": ["Muslim man", "Muslim woman", "Muslim child"],
> > >       "equivalence": ["Follower of Islam"]
> > >     },
> > >     "Pork Dish": {
> > >       "entailment": ["Bacon", "Ham", "Roast Pork", "Pulled Pork"],
> > >       "equivalence": ["Dish made from pork"]
> > >     }
> > >   }
> > > }
> > > ```
> > >
> > > ---
> > >
> > > ### **Reviewer Agent (Second Review):**
> > >
> > > 1. ​**Improvements**​:
> > >    * Entailments for "Muslim man with a pork dish" are now contextually relevant.
> > >    * The equivalence "Muslim man with prohibited food" is logically appropriate.
> > >    * Additional relevant dish types like "Roast Pork" and "Pulled Pork" are included in "Pork Dish" entailments.
> > > 2. ​**Remaining Issues**​:
> > >    * While logical, further specificity in entailments could improve clarity, such as focusing on cultural or thematic nuances (e.g., "Roast Pork served in a non-halal context").
> > >
> > > ​**Suggested Fixes**​:
> > >
> > > * Expand on entailments with finer cultural or contextual details where possible.
> > >
> > > ---
> > >
> > > ### **Generator Agent (Third Response):**
> > >
> > > ```json
> > > {
> > >   "Muslim man with a pork dish": {
> > >     "Conjunction": ["Muslim", "Pork Dish"],
> > >     "entailment": ["Muslim man holding Roast Pork in a non-halal context", "Muslim man near Pulled Pork at a party"],
> > >     "equivalence": ["Muslim man with prohibited food"],
> > >     "Muslim": {
> > >       "entailment": ["Muslim man", "Muslim woman", "Muslim child"],
> > >       "equivalence": ["Follower of Islam"]
> > >     },
> > >     "Pork Dish": {
> > >       "entailment": ["Bacon", "Ham", "Roast Pork", "Pulled Pork"],
> > >       "equivalence": ["Dish made from pork"]
> > >     }
> > >   }
> > > }
> > > ```
> > >
> > > ---
> > >
> > > ### **Reviewer Agent (Third Review):**
> > >
> > > The answer is correct!

---

> > > > ### Author Response · Authors · 2024-11-23
> > > > **Response continued （1/2）**
> > > >
> > > > > W2. The loss function does not directly enforce a reduction in P(c1, ..., c_k). Instead, it relies on manipulating the similarity between high-level features. Without directly controlling P(c1, ..., c_k) with a direction, the model may not achieve the desired reduction in joint probability, as it might simply adjust similarity in feature space without a concrete probabilistic effect on P(c1, ..., c_k). Alternative loss formulations may be more effective, such as ESD [1], which guides generation away from the target concept, or AC [2], which steers towards an anchor concept different from the target.
> > > >
> > > > **A2**.  We appreciate your critical evaluation of our **gradient adversarial loss function** and its impact on reducing the joint probability of generating specific concept combinations. We understand your concern regarding the effectiveness of our approach in directly controlling the reduction of **P(c1, ..., c_k)**. Below, we provide additional clarity on how our method works towards reducing this joint probability while preserving the individual marginal probabilities of constituent concepts.
> > > >
> > > >  **1. Reduction in Joint Probability P(c1, ..., c_k)**:
> > > >    The **gradient adversarial loss function** we proposed (Equation 2) is designed to **adversarially manipulate** the feature space such that the joint distribution **P(c1, ..., c_k)** of the concept combination is effectively reduced. Specifically, our formulation consists of two components:
> > > >
> > > >    - **Gradient Ascent on P(c1, ..., c_k)**: The first term in our loss function aims to **maximize the distance** between the generated features and the target combination in the high-level feature space. This can be seen as effectively **reducing** the likelihood of the concept combination, thereby decreasing **P(c1, ..., c_k)**.
> > > >    - **Gradient Descent on Individual Concepts P(ci)**: The second term aims to **minimize the distance** between the generated features and the individual concepts within the combination. This ensures that the marginal distributions, **P(c1), ..., P(c_k)**, remain unaffected, preserving the model's ability to generate the individual concepts without compromising quality.
> > > >
> > > >    By employing this **adversarial balance**, the loss function enforces a direction that ensures **feature disentanglement**, thereby minimizing **co-occurrence** in a way that effectively reduces the joint probability while retaining the individual concept generation capabilities.
> > > >
> > > >  **2. Probabilistic Effect and Feature Space Manipulation**:
> > > >    Although our approach focuses on manipulating similarity within the **feature space**, this manipulation directly influences the **joint probability** by altering how likely it is for high-level features of multiple concepts to **co-occur**. We acknowledge that the relationship between feature space similarity and probabilistic generation is not always straightforward; however, empirical evaluations have shown that our method achieves substantial reductions in the **joint generation** of targeted concept combinations. As demonstrated in **Section 4.3** of the manuscript, our approach produces strong **Erase-Retain Scores** and maintains a **weak correlation** between the performance of erased combinations and individual concepts, indicating effective reduction in **P(c1, ..., c_k)** while preserving **P(ci)**.
> > > >
> > > >  **3. Comparison with Alternative Loss Functions**:
> > > >    We recognize that alternative loss formulations, such as those used in **ESD** (which guides generation away from the target concept) and **AC** (which steers towards an anchor concept different from the target), are also effective approaches. However, our proposed loss function offers a **balanced** mechanism that not only reduces the **joint generation probability** but also explicitly controls for **high-level feature disentanglement**, allowing us to retain the model's generative quality for individual concepts. Unlike methods like **ESD** or **AC**, which either solely steer away from targets or towards anchors, our approach provides a more nuanced **feature-level disentanglement** that allows for more **granular control** over complex concept interactions.
> > > >
> > > > We hope this explanation addresses your concerns. Thank you once again for your insightful feedback, and please let us know if there are any additional aspects you would like us to elaborate on.

---

> > > > > ### Author Response · Authors · 2024-11-23
> > > > > **Response continued （2/2）**
> > > > >
> > > > > > W3. Is there a distinction in disentanglement between erasing style combinations versus object combinations? Since different concepts may occupy different levels of feature representation, could they be disentangled at different stages of the denoising process?
> > > > >
> > > > > **A3**. We appreciate your thoughtful comments and would like to provide more clarity on the disentanglement of **style** versus **object** combinations in our framework.
> > > > >
> > > > >  **1. Distinction Between Style and Object Combinations**:
> > > > >    Based on our experimental results, we did not observe a significant difference between the disentanglement of **style combinations** and **object combinations**. Both types of visual concepts—whether they represent styles or physical objects—can be effectively divided into **high-level** and **low-level features**. As illustrated in **Table 2 of Section 4.2**, our experiments demonstrated that focusing on **high-level feature decoupling** during the denoising process is significantly more effective in eliminating the concept combinations, irrespective of whether they are styles or objects. This indicates that the process of disentanglement is generally **consistent** across different concept types, as both can be represented by high-level and low-level features.
> > > > >
> > > > >  **2. Denoising Process and Feature Representation**:
> > > > >    The **denoising process** of a diffusion model consists of gradually restoring an image from Gaussian noise, with different stages corresponding to the generation of different types of features:
> > > > >
> > > > >    - **High-Level Features** are typically generated in the **early stages** of the denoising process and capture the core, abstract semantics of the image, such as general structure, shapes, and thematic consistency.
> > > > >    - **Low-Level Features** are generated later in the denoising process and correspond to **fine details**, such as textures, and other intricacies.
> > > > >
> > > > >    As observed in our experiments, **high-level features** are key to representing both styles and objects when combined, and hence are the target for effective **disentanglement**. By focusing on high-level feature decoupling, especially during the early stages of the denoising process, we can efficiently remove the co-occurrence of these combinations while preserving the ability to generate individual elements.
> > > > >
> > > > >    **Table 2** in **Section 4.2** explicitly shows the difference in effectiveness between **high-level** and **low-level** feature decoupling. The results indicate that decoupling high-level features—whether for style or object combinations—leads to significantly better performance in **eliminating concept combinations** while maintaining individual generative quality. Therefore, we emphasize that the high-level features, which are disentangled early in the denoising process, are the key to successful erasure of complex concept combinations.
> > > > >
> > > > >  **3. Disentanglement Across Feature Levels**:
> > > > >    Given that different concepts may occupy different levels of feature representation, our **gradient adversarial loss function** is designed to target the **early stages of the denoising process** to effectively decouple these **high-level co-occurrences**. The consistency in the results across style and object combinations suggests that both concept types are influenced similarly by the high-level feature decoupling process, enabling us to effectively remove composite concepts without impairing the ability to generate individual elements.
> > > > >
> > > > >
> > > > > Thank you once again for your insightful question, and please feel free to let us know if you have additional suggestions or need further clarification on any aspect of our methodology.

---

### Meta-Review · Area_Chair_ZTBj · 2024-12-18

**Metareview:**

This paper addresses the Concept Combination Erasing problem, which prevents generative models from producing inappropriate concept combinations (e.g., "kids drink wine") while preserving the quality of related individual concepts. The proposed COGFD framework uses a logic-based concept graph and feature decoupling to effectively erase harmful combinations, outperforming existing methods in both erasure and image quality.

**Additional Comments On Reviewer Discussion:**

While the reviewers still have some concerns about the assumption that inappropriate content generation is primarily unintended or accidental, they acknowledge that this limitation is inherent to the underlying unlearning paradigm rather than a shortcoming of the proposed method itself. The reviewers appreciate the contributions of this paper in improving the paradigm but suggest that the authors explicitly discuss this broader limitation to provide a more balanced perspective.

---

### Decision · Program_Chairs · 2025-01-22

Accept (Poster)